# Duo Emission of CVD Nanodiamonds Doped by SiV and GeV Color Centers: Effects of Growth Conditions

**DOI:** 10.3390/ma15103589

**Published:** 2022-05-18

**Authors:** Kirill V. Bogdanov, Mikhail A. Baranov, Nikolay A. Feoktistov, Ilya E. Kaliya, Valery G. Golubev, Sergey A. Grudinkin, Alexander V. Baranov

**Affiliations:** 1Center of Information and Optical Technologies, ITMO University, Kronverksky Pr. 49, bldg. A, 197101 St. Petersburg, Russia; kirw.bog@gmail.com (K.V.B.); mbaranov@mail.ru (M.A.B.); kaliyailya2802@gmail.com (I.E.K.); grudink@gvg.ioffe.ru (S.A.G.); 2Ioffe Institute, Polytechnicheskaya 26, 194021 St. Petersburg, Russia; feokt@gvg.ioffe.ru (N.A.F.); golubev@gvg.ioffe.ru (V.G.G.)

**Keywords:** nanodiamonds, color centers, chemical vapor deposition, photoluminescence, Raman spectroscopy

## Abstract

The investigation of the hot filament chemical vapor deposition nanodiamonds with simultaneously embedded luminescent GeV^−^ and SiV^−^ color centers from solid sources showed that both the absolute and relative intensities of their zero-phonon lines (at 602 and 738 nm) depend on nanodiamond growth conditions (a methane concentration in the CH_4_/H_2_ gas mixture, growth temperature, and time). It is shown that a controlled choice of parameters of hot filament chemical vapor deposition synthesis makes it possible to select the optimal synthesis conditions for tailoring bicolor fluorescence nanodiamond labels for imaging biological systems.

## 1. Introduction

Diamond nanoparticles (nanodiamonds, ND) with embedded fluorescent color centers, being biocompatible non-toxic materials that exhibit stable emission with high brightness without photobleaching, are considered as promising fluorescence labels for imaging biological systems [1,2]. Importantly, methods to chemically modify the surface of NDs that allow entering cells, targeting labeling, sensing, and drug delivery, are described [3]. Except for NDs with well-studied NV^−^ color centers, other fluorescent color centers based on group IV elements, such as the negatively charged silicon vacancy (SiV^−^) and germanium vacancy (GeV^−^), are attracting much attention [4,5,6,7]. They advantageously demonstrate emission mostly in their narrow zero-phonon line (ZPL) with a much better spectral stability than the NV^−^ centers, even under ambient conditions and a much larger Debye–Waller factor (up to ~0.7 for SiV^−^ and ~0.6 for GeV^−^ [8,9,10]) due to weak electron–phonon coupling [11]. With ZPL at 738 and 602 nm for SiV^−^ and GeV^−^, respectively, they may be considered as good candidates for multicolor biolabeling [2].

The SiV^−^ and GeV^−^ centers have a very similar electronic level structure with D_3d_ symmetry [12]. Both negatively charged silicon-vacancy (SiV^−^) and germanium-vacancy (GeV^−^) centers can be formed with several techniques: ion implantation [13,14], shock wave synthesis, and doping during HPHT [15,16] and chemical vapor deposition (CVD) growth [17,18,19,20]. Among these methods, the CVD approaches look most attractive, since the ion implantation often results in lattice damage, causing strains and the degradation of optical properties. For the detonation and HTHP syntheses, it is hard to provide the reliable production of a high-purity material with a controlled amount of color centers and good optical properties, and they cannot provide mass production of fluorescent ND. On the other hand, according to [20], the SiV^−^ or GeV^−^ centers possessing bright emissions can be successfully formed in NDs by CVD synthesis using small pieces of crystalline Si or Ge placed on the holder near the NDs as a source of ions. This approach opens a way to create a multicolor solid-state source of light for bioimaging by using diamond nanoparticles with simultaneously or subsequently embedded several fluorescent color centers, e.g., the SiV^−^ and GeV^−^ centers emitting at 738 and 602 nm. The hot filament chemical vapor deposition (HFCVD) method was used in [19], allowing the synthesis of NDs with simultaneously introduced GeV^−^ and SiV^−^ color centers. The known data on the measurement of the luminescence excitation spectra of SiV^−^ and GeV^−^ centers [7] show that the luminescence of both centers can be effectively excited by radiation of one wavelength, for example, 488 nm yielding a two-color emitter of bright and spectrally narrow lines. Using this approach, however, the problem of controlling the level of doping of the ND crystal with color centers of different types arises, which determines the spectral and kinetic parameters of the fluorescence of the centers and the possible effects of the interaction of the centers with each other. As was shown in [19,20,21,22], the luminescence intensity of color centers in CVD diamonds depends strongly on the growth conditions. Therefore, it becomes urgent to study the influence of the growth conditions of NDs doped with different centers and post-synthesis treatment of the ND on optical parameters of luminescence of the embedded color centers.

In this work, we report the HFCVD nanodiamonds with simultaneously embedded luminescent GeV^−^ and SiV^−^ color centers from solid sources and show that both the absolute and relative intensities of their ZPLs (at 602 and 738 nm) depend on ND growth conditions (a methane concentration in the CH_4_/H_2_ gas mixture, growth time, and substrate temperature).

## 2. Materials and Methods

### 2.1. The Fabrication of Luminescent Nanodiamonds

The formation of nanodiamonds doped with luminescent GeV^−^ and SiV^−^ centers was carried out in two stages. First, the method of aerosol deposition of nanodiamonds of detonation synthesis with a size of ~5 nm on a silicon substrate, was used to form individual isolated centers of nucleation of diamond particles on the surface for subsequent CVD synthesis with a concentration of 10^7^ cm^−2^ [23]. Next, the samples of the substrate with deposited nucleation centers, which were previously subjected to atomic hydrogen etching to activate the surface, were used to grow diamond particles by HFCV from hydrogen–methane mixture. This method is based on the decomposition of methane and hydrogen near the hot helix into hydrocarbon radicals and atomic hydrogen, which diffuse to the heated substrate, leading to the growth of diamond particles on pre-deposited diamond nucleation centers [19] (see Figure 1).

The formation of color centers during the growth of diamond particles was carried out by introducing a dopant into the gas phase. A schematic diagram of doping of diamond particles during HFCVD growth is shown in Figure 1. The substitutional donors were volatile GeH_x_ and SiH_x_ radicals obtained with atomic hydrogen etching bulk crystalline germanium and silicon wafers that were located nearby on the substrate holder during CVD growth and used as a solid-state source of Ge and Si atoms. The temperatures of the solid-state sources were higher than the silicon substrate temperature by ~130 °C. The volatile GeH_x_ and SiH_x_ radicals move to the substrate surface by means of the diffusion process. The Ge and Si atoms, integrating into the diamond lattice from the gas phase, promoted the formation of ensembles of GeV^−^ and SiV^−^ centers in ND. The investigated samples of luminescent NDs were obtained under the following parameters of the HFCVD process: temperature of the tungsten filament, 2000–2200 °C; substrate temperature, T = 600–750 °C; operating pressure in the reactor—50 Torr; hydrogen flow—500 sccm; a methane concentration in the CH_4_/H_2_ gas mixture—C(CH_4_) = 1–8%; the growth time of diamond particles is t = 2–4 h. The distance between the tungsten spiral and the substrate holder is 10–12 mm. The size of the substrate is 2 mm × 2 mm. Before starting the HFCVD growth process, the reactor is pumped down to a pressure of 10^−2^ Torr.

### 2.2. Scanning Electron Microscopy, Photoluminescence, and Raman Setup

The typical scanning electron microscope (SEM) images of nanodiamonds particles with embedded luminescent GeV^−^ and SiV^−^ color centers formed at different growth parameters. The images were obtained with the Zeiss Scanning Electron Microscope, “Merlin”, at an accelerating voltage of 10 kV at a probe current of 150 pA. To improve the image quality and topological contrast, the samples were fixed with carbon tape to create a conductive bridge between the silicon substrate and the sample holder, and the signals from the InLens and Everhart-Thornley SE2 detectors were recorded simultaneously.

The luminescence spectra of the nanodiamonds were measured using a Renishaw “InVia” Raman spectrometer equipped with a confocal microscope, liquid-nitrogen-cooled CCD, and 1800 lines/mm grating. The spectral resolution of the spectrometer was ~2 cm^−1^. The excitation laser radiation of 488 nm was focused by a 100× lens (NA = 0.9) into a spot with a diameter of ~2 μm on the single selected ND crystal. To correctly compare the luminescence intensities of GeV^−^ and SiV^−^ centers that possess different emission wavelengths, they were normalized on the spectral sensitivity of the spectrometer determined preliminary by a standard “black body” emission. Since the used spectrometer allows both the luminescence and Raman spectra to be obtained simultaneously from the same individual ND, for a comparison of the luminescence intensities obtained from nanocrystals of different sizes, the luminescence intensities were also normalized to the intensity of the diamond Raman line of ~1332 cm^−1^ (521.9 nm), which is proportional to the illuminated nanocrystal volume. All measurements were carried out at least 5 times to prove the reproducibility of the data obtained.

## 3. Results and Discussion

Figure 2a shows the luminescence (PL) spectra of HFCVD NDs with simultaneously embedded GeV^−^ and SiV^−^ centers, normalized to the intensity of the diamond Raman line, at different concentrations of methane in the hydrogen–methane mixture from 1% to 6%. ND particles were grown for 2h on a crystalline silicon substrate heated up to 720 °C. The synthesis time and substrate temperature here are in the range corresponding, as will be seen below, to the conditions for simultaneous observation of the ZPLs of both GeV^−^ and SiV^−^ centers. The luminescence spectra clearly demonstrate that not only absolute but also relative intensities of ZPLs of the GeV^−^ and SiV^−^ centers at 602 nm and 738.2 nm depend on the methane concentration in the mixture, as shown in the inset of Figure 2. Figure 1b displays the 2D PL mapping of the NDs obtained at a methane concentration of 6%. It can be seen the absolute intensities of ZPLs vary for different particles. The relative intensities of ZPLs are practically the same for NDs.

At a low concentration of methane in the hydrogen–methane mixture, only one fairly intense ZPL from SiV^−^ centers is observed in the spectra, while the ZPL from GeV^−^ centers is almost 30 times less intense. As the methane concentration increases, the SiV^−^ luminescence intensity reaches a maximum at a methane concentration of 2% and then begins to decrease by more than an order of magnitude at a methane concentration of 6%. On the other hand, an increase in methane concentration leads to a monotonic increase in the intensity of ZPL of GeV^−^ centers. It can also be seen that an increase in the methane concentration leads to an increase in the intensity of the broadband background in the region of 550–700 nm, which, according to [19,20], is associated with an increase in the volume of surface regions containing structural defects and sp^2^-hybridized carbon. The Raman spectra presented in Figure 2c demonstrate a characteristic diamond band at 1332 cm^−1^, the width of which increases with increasing methane concentration in the mixture, indicating an increase in the defectiveness of the crystal structure and the presence of the sp^2^ carbon phase in the synthesized ND.

It is thermodynamically favorably for part of silicon atoms to be located at the grain boundaries, and these silicon atoms do not form color centers SiV^−^ and are inactive in photoluminescence [24]. As the methane concentration increases, the number of grain boundaries increases, and as a result, the number of SiV^−^ color centers decreases. The appearance of structural defects is most likely associated with the quasi-nonequilibrium rapid growth of the surface layer of nanocrystals with an increase in the number of carbon atoms in the reaction gas mixture. Regarding the increase in the size of NDs and the distortion of their faces, a growth of a number of grain boundaries with increasing the methane concentration is clearly seen in their SEM images shown in Figure 2b. The simultaneous increase in the intensity of ZPL GeV^−^ centers may indicate that GeV^−^ centers are localized to a greater extent in the more defective near-surface layer of diamond nanocrystals, while the SiV^−^ centers are mainly localized in the nanocrystal area with the more ordered crystal structure. We can propose a scenario for an explanation of the differences between the dependences of the ZPL intensities of the SiV^−^ and GeV^−^ color centers on the methane concentration. An increase in the methane concentration leads to an increase in the defect-induced strain distribution in a lattice. In some regions of NDs, the strains could result in an expanded crystal lattice that stimulates the penetration of impurity atoms into the lattice and the formation of color centers. At low methane concentrations, silicon atoms are more likely to penetrate due to their smaller size than germanium atoms. As the methane concentration increases, the number of germanium atoms in the lattice and the number of GeV^−^ centers also increase. Apparently, the probability of capture of vacancies by germanium atoms is higher than that of silicon atoms, which explains the increase in the GeV^−^ ZPL intensity with respect to the SiV^−^ ZPL intensity. Finally, the comparable ZPL intensities of the GeV^−^ and SiV^−^ centers that are most attractive for realizing multicolour light sources were found for HFCVD NDs synthesized at a methane concentration ranging from 4% to 5% in the gas mixture.

Figure 3 shows the luminescence spectra of NDs with embedded GeV^−^ and SiV^−^ centers synthesized at different substrate temperatures of 630 °C, 700 °C, and 750 °C. Samples were grown for 2 h at a methane concentration of 4%.

Our studies have made it possible to determine the substrate temperature range at which both luminescent centers can form in HFCVD NDs. It is shown that diamond particles on a silicon substrate begin to grow at temperatures above 570 °C. As the temperature increases, a monotonic increase in the ZPL intensity of SiV^−^ centers is observed. The intensity of ZPL GeV^−^ centers also increases, reaches an extremum in the region of ~700 °C, and noticeably decreases with a further increase in temperature. In [21], such a temperature dependence of the intensity of ZPL GeV^−^ centers in HFCVD NDs was associated with an increase in the mobility of deposited germanium radicals (atoms), sufficient for their involvement in the formation of germanium nanocrystals on the surface of NDs. As a result, at high substrate temperatures, a significant part of germanium atoms does not participate in the formation of GeV^−^ centers, which leads to a low intensity of ZPL GeV^−^ centers. Thus, the substrate temperature in the region of 700 °C can be considered close to optimal for the proposed HFCVD formation of bicolor luminescent NDs doped GeV^−^ and SiV^−^ color centers.

Figure 4 shows the luminescence spectra of diamond particles formed at different growth times (2, 3, 4 h) at a substrate temperature of 700 °C and a methane concentration of 4%. An increase in the growth time leads to a two-fold increase in the average size of diamond particles from 800 nm to 1500 nm (Figure 4). At the same time, the parameters of the ZPLs, both GeV^−^ and SiV^−^ centers in the ND luminescence spectra, are changed but not as strong as may be expected. We see an approximately two-fold increase in the GeV^−^ ZPL intensity with an analogous reduction of the SiV^−^ ZPL with the growth of the ND sizes. Since the spectra were normalized to the diamond Raman band intensity, it is not a simple ND volume effect. It is very likely that the outstripping growth of GeV^−^ ZPL intensity is associated with an increase in the volume of ND regions containing structural defects, as was noted in the discussion of the effect of the methane concentration in the reaction mixture on the GeV^−^ ZPL intensity. 

Thus, the variation in the synthesis time of HFCVD NDs in the range of 2–4 h at the optimized substrate temperature and methane concentration in the methane–hydrogen mixture does not lead to a noticeable deterioration in the luminescence parameters of the ZPLs of the GeV^−^ and SiV^−^ centers at 602 nm and 738.2 nm.

## 4. Conclusions

The luminescence spectra of individual nanodiamonds with simultaneously embedded GeV^−^ and SiV^−^ color centers synthesized by the hot filament chemical vapor deposition method under different synthesis conditions have been studied. The formation of color centers occurs due to the introduction of a dopant from solid sources into the gas phase and then into the diamond lattice. The spectra demonstrate two narrow ZPLs of GeV^−^ and SiV^−^ color centers at 602 nm and 738.2 nm, respectively, whose intensities strongly depend on the synthesis conditions.

The analysis of measured luminescence spectra demonstrates that not only absolute but also relative intensities of ZPLs of the GeV^−^ and SiV^−^ centers depend on the methane concentration in the mixture. The optimal concentration of methane in the range from 4% to 5% was determined, which corresponds to the growth of bicolor nanodiamonds with comparable ZPL intensities of the GeV^−^ and SiV^−^ centers. The study of the substrate temperature dependences of photoluminescence showed that the ZPL intensity of the SiV^−^ center increases with temperature and the ZPL intensity of the GeV^−^ center is maximal at the temperature of ~700 °C. It was found that an increase in NDs size leads to an approximately two-fold increase in the GeV^−^ ZPL intensity with an analogous reduction of the SiV^−^ ZPL.

The analysis of the experimental results allowed us to conclude that GeV^−^ centers are localized in the more defective regions of diamond nanocrystals, while SiV^−^ centers are mainly localized in the nanocrystal region with the more ordered crystal structure. An increase in methane concentration, substrate temperature, and growth time results in increasing the volume of ND regions containing structural defects and, consequently, the growth of ZPL intensity in GeV^−^ centers, and a reduction of ZPL in SiV^−^ centers.

The dependences obtained make it possible to select the parameters for the synthesis of nanodiamonds (substrate temperature of ~630–700 °C, methane concentration of ~4–5%, the growth time of 2–4 h) to achieve the optimal luminescent responses of both GeV^−^ and SiV^−^ color centers embedded in the nanodiamonds. Thus, the possibility of creating stable two-color narrow-band emitters by simultaneous doping of CVD nanodiamonds with silicon and germanium atoms, with the possibility of varying and redistributing the intensities of luminescent responses by changing the synthesis parameters, has been demonstrated.

## Figures and Tables

**Figure 1 materials-15-03589-f001:**
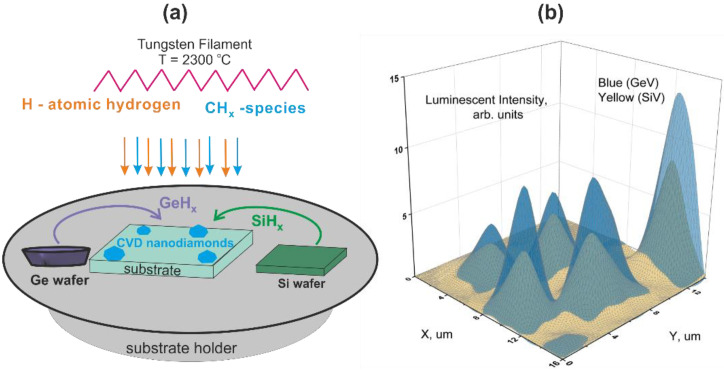
(**a**) A schematic diagram of growth and doping of diamond particles by Ge and Si atoms during hot filament chemical vapor deposition; (**b**) 2D ZPL intensity mapping of SiV^−^ or GeV^−^ centers at a methane concentration of 5%.

**Figure 2 materials-15-03589-f002:**
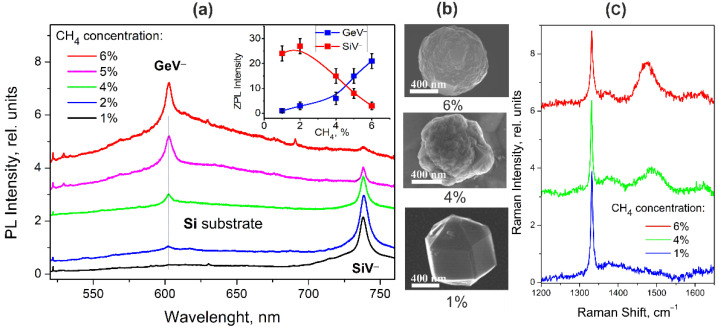
(**a**) The luminescence spectra of HFCVD NDs with embedded GeV^−^ and SiV^−^ centers, normalized to the intensity of the diamond Raman line, grown at various methane concentrations in the hydrogen–methane mixture from 1% to 6% during 2 h on a crystalline silicon substrate heated up to 720 °C. The inset shows the dependences of the integrated ZPL intensities of the GeV^−^ and SiV^−^ centers on the methane concentration in the mixture; (**b**) typical SEM images of ND particles grown at various CH_4_ concentrations:(**c**) typical Raman spectra for a series of ND samples synthesized with 1%, 4%, and 6% contents of CH_4_ in the gas mixture, demonstrating characteristic diamond band at 1332 cm^−1^.

**Figure 3 materials-15-03589-f003:**
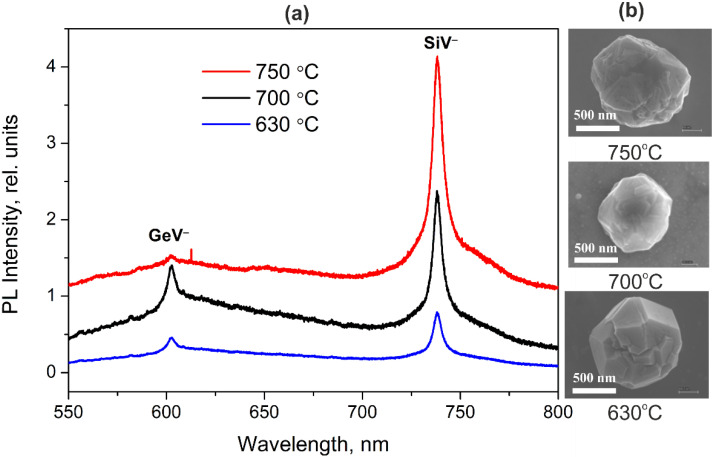
(**a**) Luminescence spectra of HFCVD NDs with embedded GeV^−^ and SiV^−^ centers synthesized on silicon substrate at different substrate temperatures of 630 °C, 700 °C, and 750 °C; (**b**) typical SEM images of ND particles grown at different substrate temperatures.

**Figure 4 materials-15-03589-f004:**
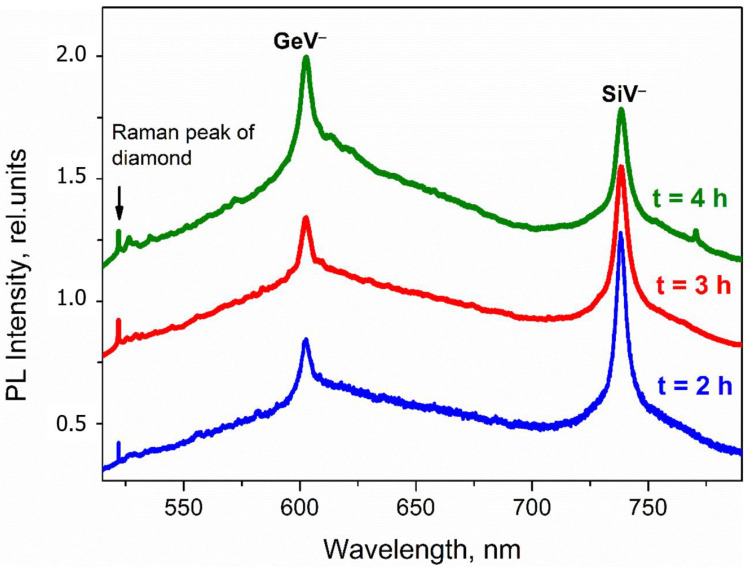
Photoluminescence spectra of nanodiamonds obtained over a growth time of 2, 3, 4 h. The substrate temperature—700 °C; the methane concentration—4%.

## Data Availability

Not applicable.

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
