# Peer review of "Duo Emission of CVD Nanodiamonds Doped by SiV and GeV Color Centers: Effects of Growth Conditions"

_materials, 2022, doi:10.3390/ma15103589_

Round 1

Reviewer 1 Report

  1. In the study of 'Luminescence spectra of HFCVD NDs with embedded GeV and SiV centers synthe- 178 sized on silicon substrate at different substrate temperatures',  why only  630 °С, 700 °С, and 750 °С were considered? why not any other temp. values? Any physical significance for considering only 630 °С, 700 °С, and 750 °С.
  2.  As per author's statement, 'samples were grown during 2 hours at a methane concentration of 4%'. How to ensure that the specified duration and concentration are sufficient for the sample study? Any specific reason? or any reference for that?
  3. If possible, add more contents under introduction part to support your study.

Author Response

We are grateful to the Reviewer for careful reading of our manuscript and useful recommendations and criticism. We have carefully analyzed and corrected our manuscript in accordance with the referee’s remarks.

  1. In the study of 'Luminescence spectra of HFCVD NDs with embedded GeV and SiV centers synthesized on silicon substrate at different substrate temperatures', why only  630 °С, 700 °С, and 750 °С were considered? why not any other temp. values? Any physical significance for considering only 630 °С, 700 °С, and 750 °С.

Response 1.

As it was mentioned in the manuscript, both GeV- and SiV- ZPL are reliably observed simultaneously in the temperature range from 5700C to 8000C. The temperature values of 630 0C, 700 0C, and 750 0C were chosen since the related PL spectra reflect the main important features of the temperature-dependent evolution of the GeV and SiV ZPL intensity.

  1.  As per author's statement, 'samples were grown during 2 hours at a methane. How to ensure that the specified duration and concentration are sufficient for the sample study? Any specific reason? or any reference for that?

Response 2.

The duration of the ND growth of 2 hours and the methane concentration of 4% were chosen because at these parameters the ZPL intensities of the GeV and SiV are comparable in the emission spectra of NDs.

  1. If possible, add more contents under introduction part to support your study.

Response 3.

To clarify this topic, we added the phrase and reference in the text “As was shown in [19- 22] the luminescence intensity of color centers in CVD diamonds depend strongly on the growth conditions.” 

Reviewer 2 Report

In the experimental part, was any method used to de-agglomerate the ND particles on the substrate?

Was the surface temperature of the solid state sources (Ge and Si) the same as the substrate temperature?

How thick is the diamond film? Or under these conditions, did a continuous film not form on the substrate?

The  word “figure 3” is not insert in the text

Is it possible to estimate the concentration of GeV and SiV centers?

Author Response

We are grateful to the Reviewer for careful reading of our manuscript and useful recommendations and criticism. We have carefully analyzed and corrected our manuscript in accordance with the referee’s remarks.

In the experimental part, was any method used to de-agglomerate the ND particles on the substrate?

Response 1.

The use of the aerosol spraying method [21] makes it possible to deposit individual isolated nanodiamonds of detonation synthesis on substrates at a rather low surface concentration of 107 cm–2, to avoid nanoparticle agglomeration.

Was the surface temperature of the solid state sources (Ge and Si) the same as the substrate temperature?

Response 2.

No, the temperatures of the solid state sources were higher than the silicon substrate temperature by ca. 130 0C.

In the text: The temperatures of the solid state sources were higher than the silicon substrate temperature by ~130 0C.

How thick is the diamond film? Or under these conditions, did a continuous film not form on the substrate?

Response 3.

At these pretreatment and growth conditions, the continuous diamond films are not formed on the substrate.

The word “figure 3” is not insert in the text.

Response 4.

This misprint was corrected (Figure 4 has been replaced by Figure 3)

Is it possible to estimate the concentration of GeV and SiV centers?

Response 5.

Without a doubt, knowledge of the concentrations of color centers in ND is very important. However, for nanodiamonds, determining the concentration is a rather difficult task. It is known that EPR spectroscopy methods can be used to determine the concentration of NV centers in ND [sее е.g. Bogdanov KV, APL Materials 6, 086104 (2018); https://doi.org/10.1063/1.5045535]. However, to date, we are unaware of works that would offer reliable methods for determining the content of GeV or SiV color centers in diamond particles.

Reviewer 3 Report

It is interesting to co-incorporate Si and Ge into the nanodiamonds to form SiV and GeV color centers by HFCVD, and it is also useful to study the basic influence of the growth parameters on the emission of both the color centers. It is suggested that this manuscript need some revisions before it is suitable for publication.

Besides the studied parameters, the influence of the pressure is also required.

The detailed concentration of the doped Si and Ge should be measured.

For the doping process by using the solid wafers, how to control the flow of the evaporated atoms? Why can they move to the substrate surface? By flowing or diffusions or some others?

Some basic characterizations on the nanodiamonds should be provided, such as the Raman.

Will the silicon substrate also induce the silicon doping in the nanodiamonds? Please provide some comparative experiment for proving it (growing nanodiamonds without placing the Si wafer in).

Author Response

We are grateful to the Reviewer for careful reading of our manuscript and for useful recommendations and criticism. We have carefully analyzed and corrected our manuscript in accordance with the referee’s remarks.

Besides the studied parameters, the influence of the pressure is also required.

Response 1.

We agree with the reviewer that hydrogen pressure is an important parameter of the CVD growth of ND because the magnitude of pressure affects the crystalline perfection of nanodiamonds [see for example Fabisiak, K.; Torz-Piotrowska, R.; Staryga, E.; Szybowicz, M.; Paprocki, K.; Banaszak, A.; Popielarski, P. The Influence of Working Gas on CVD Diamond Quality. Materials Science and Engineering: B 2012, 177, 1352–1357. https://doi.org/10.1016/j.mseb.2011.12.013 ]. In our work, a pressure of 50 Torr was chosen, as indicated in the experimental section of the manuscript, at which an acceptable quality of the crystal structure of nanodiamonds is achieved in the setup used.

The detailed concentration of the doped Si and Ge should be measured.

Response 2.

In this work, the concentrations of the doped Si and Ge atoms have not been measured since we believe that in this case, it is rather the concentrations of the corresponding color centers are important. We are currently investigating the possible determination of the content of SiV- and GeV- centers in synthesized CVD ND using EPR spectroscopy.

For the doping process by using the solid wafers, how to control the flow of the evaporated atoms?

Response3:

Control of the flow of the evaporated atoms in the case of using the solid wafers is a really hard problem for the CVD growth techniques. It can be done only qualitatively by controlling the temperature of the solid wafer. In our experiments, the temperature of the wafers was determined by filament temperature and the distance between filament and substrate holder. In experiments on methane concentration and growth time dependencies these parameters and, consequently the value of the flow, were maintained constant.

Why can they move to the substrate surface? By flowing or diffusions or some others?

Response 4.

We clarify this topic

In the text:  The volatile GeHx and SiHx radicals move to the substrate surface by means of the diffusion process.

Some basic characterizations on the nanodiamonds should be provided, such as the Raman.

Response 5.

Typical Raman spectra for a series of ND samples synthesized with different concentrations of methane in the gas mixture are added to Figure 2c in the corrected manuscript. An appropriate text and caption are added to the manuscript and caption to Figure 2.

In the text: The Raman spectra presented in Figure 2c  demonstrate a characteristic diamond band at 1332 cm–1, the width of which increases with increasing methane concentration in the mixture, indicating an increase in the defectiveness of the crystal structure and presence of the sp2 carbon phase in the synthesized ND.

Will the silicon substrate also induce the silicon doping in the nanodiamonds? Please provide some comparative experiments for proving it (growing nanodiamonds without placing the Si wafer in).

Response 6.

Since, as mentioned in the revised manuscript, the substrate temperature is 1300C lower than the temperature of the solid state wafer, we believe that its contribution to ND doping most likely can be neglected.

Reviewer 4 Report

In the present paper, the authors prepared the doped nanodiamonds at various conditions and examined their luminescence. The overall actuality of this work is high, given the potential applications of such an emission, however, its novelty, the level, at which it is conducted, and the poverty of the used techniques does not make it possible to publish this paper in Materials. This paper might be suitable for another journal, such as Inorganics, after the revisions.

1) Additional characterization of the obtained nanodiamonds is needed, such as Raman spectroscopy (under a NIR laser to avoid luminescence), PXRD, etc.

2) Why was the luminescence measured using a Raman spectrometer? The obtained intensity correlates with various parameters, including the sample crystallinity, while in reality the intensity dependence may be quite different, and this is what a spectrometer can demonstrate. Such measurement should be run additionally.

3) The conclusions do not conclude anything. They claim that the properties depend on the preparation conditions, and the dependence allows to select the optimal parameters. The analysis of the obtained results would be done. What is the dependence and what is its origin? Which are those optimal conditions? Etc.  

Author Response

We are grateful to the Reviewer for careful reading of our manuscript and useful recommendations and criticism. We have carefully analyzed and corrected our manuscript in accordance with the referee’s remarks.

  • Additional characterization of the obtained nanodiamonds is needed, such as Raman spectroscopy (under a NIR laser to avoid luminescence), PXRD, etc.

Response 1.

We agree with the referee. Typical Raman spectra for a series of ND samples synthesized with different concentrations of methane in the gas mixture are added to Figure 2 in the corrected manuscript. The Raman spectra were excited by cw radiation of 488 nm. In this case, ND luminescence under of interest does not interfere with the Raman spectra as can be seen in Figure 4. As for the NIR excitation, we would like to pay the referee attention that a diamond Raman cross-section decreases strongly in the NIR region making detection of the Raman signals from the single ND problematic. The Raman spectra presented in Figure 2c show a characteristic diamond band at 1332 cm–1, the width of which increases with increasing methane concentration in the mixture, indicating an increase in the defectiveness of the crystal structure and the presence of the sp2 carbon phase in the synthesized NDs. Relevant added information on recording Raman spectra is given in Section 2.2. At the same time, its name was changed to "Scanning Electron Microscopy, Photoluminescence and Raman setup"

In the text: The Raman spectra presented in Figure 2c demonstrate a characteristic diamond band at 1332 cm–1, the width of which increases with increasing methane concentration in the mixture, indicating an increase in the defectiveness of the crystal structure and presence of the sp2 carbon phase in the synthesized ND.

  • Why was the luminescence measured using a Raman spectrometer? The obtained intensity correlates with various parameters, including the sample crystallinity, while in reality the intensity dependence may be quite different, and this is what a spectrometer can demonstrate. Such measurement should be run additionally.

Response 2.

We cannot agree with the Reviewer. Reasons for using the Renishaw micro-Raman spectrometer inVia are that the sensitivity and spectral resolution of the spectrometer allows simultaneous measurement of luminescent and Raman spectra of a single sub-micron ND. It is necessary for a quantitative comparison of the ZPL intensities for ND crystals of different sizes. From our point of view, section 2.2 “Scanning Electron Microscopy, Photoluminescence, and Raman setup” contain enough data to answer the Reviewer remark.

  • The conclusions do not conclude anything. They claim that the properties depend on the preparation conditions, and the dependence allows to select the optimal parameters. The analysis of the obtained results would be done. What is the dependence and what is its origin? Which are those optimal conditions? Etc.  

Response 3.

To satisfy the reviewer's comments, we have added additional text to the Conclusion section that we hope will make the results clear to understand.

In the text:

The analysis of measured luminescence spectra clearly demonstrates that not only absolute but also relative intensities of ZPLs of the GeV and SiV centers at 602 nm and 738.2 nm depend on the methane concentration in the mixture. There is an optimal methane concentration at which emission of the GeV– and SiV centers with comparable ZPLs intensities takes place. The obtained data may indicate that GeV centers are localized in the more defective near-surface layer of diamond nanocrystals while SiV– centers are mainly localized in the nanocrystal region with the more ordered crystal structure. Analysis of the temperature dependences of the GeV– and SiV– ZPL intensities obtained at a practically optimal concentration of methane in a methane–hydrogen mixture shows that SiV– ZPL intensity increases with temperature. The ZPL intensity of GeV center is maximal at the temperature of ~700 0C and quickly decreases with growing temperature most likely due to the formation of Ge nanocrystals on the ND surface. As for the dependence of the GeV and SiVZPL intensities on the growth time, we can conclude that increasing the time results in an increase in the volume of ND regions containing structural defects with GeV centers.

Round 2

Reviewer 3 Report

The authors have responded to all my questions. The present manuscript can be accepted.

Author Response

We are grateful to the Reviewer for the remarks that allowed us to improve the manuscript and for the decision to accept the manuscript for publication.

Reviewer 4 Report

The authors have mainly addressed my points, though the Conclusions sections still has to be significantly re-written.

First, after being corrected, it now takes ca. 1 page - this is a retelling, not a conclusion. It should be significantly shortened.

Second, despite authors tried to finally analyze their results, the efforts are not enough. See, i.e.:

"There is an optimal methane concentration..." - which one?

"The obtained data may indicate that..." - the "may" word is not for conclusions 

"The dependences obtained make it possible to select the parameters..."  - were they selected?

etc.

Author Response

We are grateful to the Reviewer for your remarks and suggestions.  The conclusion section has been modified and shortened.  We have outlined with greater clarity and certainty the conclusions made in the work. English language and style have been corrected.

In the text:

The luminescence spectra of individual nanodiamonds with simultaneously embedded GeV and SiV color centers synthesized by the Hot Filament Chemical Vapor Deposition method under different synthesis conditions have been studied. The formation of color centers occurs due to the introduction of a dopant from solid sources into the gas phase and then into the diamond lattice. The spectra demonstrate two narrow ZPL of GeV and SiV color centers at 602 nm and 738.2 nm, respectively, whose intensities strongly depend on the synthesis conditions.

The analysis of measured luminescence spectra clearly demonstrates that not only absolute but also relative intensities of ZPLs of the GeV and SiV centers depend on the methane concentration in the mixture. The optimal concentration of methane in the range from 4% to 5 % was determined, which corresponds to the growth of bicolor nanodiamonds with comparable ZPLs intensities of the GeV and SiV centers. The study of the substrate temperature dependences of photoluminescence showed that ZPL intensity of SiV center increases with temperature and  ZPL intensity of GeV center is maximal at the temperature of ~700 0C.  It was found, that an increase in NDs size leads to an approximately two-fold increase of the GeV ZPL intensity with analogous reduction of the SiV ZPL.

The analysis of the experimental results allowed us to conclude that GeV centers are localized in the more defective regions of diamond nanocrystals while SiV centers are mainly localized in the nanocrystal region with the more ordered crystal structure.  An increase in methane concentration, substrate temperature, and growth time results in increasing the volume of ND regions containing structural defects and, consequently, growth of ZPL intensity of GeV center, and reduction of ZPL of SiV centers.

The dependences obtained make it possible to select the parameters for the synthesis of nanodiamonds (substrate temperature of ~630-700 C, methane concentration of ~4% - 5 %, the growth time of 2-4 h) to achieve the optimal luminescent responses of both GeV and SiV color centers embedded in the nanodiamonds. Thus, the possibility of creating stable two-color narrow-band emitters by simultaneous doping of CVD nanodiamonds with silicon and germanium atoms with the possibility of varying and redistributing the intensities of luminescent responses by changing the synthesis parameters has been demonstrated.
